# A Numerical Modelling Framework for Investigating the Ballistic Performance of Bio-Inspired Body Armours

**DOI:** 10.3390/biomimetics8020195

**Published:** 2023-05-08

**Authors:** Abdallah Ghazlan, Tuan Ngo, Ping Tan, Phuong Tran, Yi Min Xie

**Affiliations:** 1Department of Infrastructure Engineering, Faculty of Engineering and Information Technology, University of Melbourne, Parkville, VIC 3052, Australia; 2Defense Science and Technology Group, Edinburgh, SA 5111, Australia; 3Department of Civil and Infrastructure Engineering, School of Engineering, RMIT University, Melbourne, VIC 3000, Australia

**Keywords:** ballistic protection, bio-inspired, body armour, nacre, conch, fish scale, crustacean

## Abstract

Biological structures possess excellent damage tolerance, which makes them attractive for ballistic protection applications. This paper develops a finite element modelling framework to investigate the performance of several biological structures that are most relevant for ballistic protection, including nacre, conch, fish scales, and crustacean exoskeleton. Finite element simulations were conducted to determine the geometric parameters of the bio-inspired structures that can survive projectile impact. The performances of the bio-inspired panels were benchmarked against a monolithic panel with the same 4.5 mm overall thickness and projectile impact condition. It was found that the biomimetic panels that were considered possessed better multi-hit resistant capabilities compared to the selected monolithic panel. Certain configurations arrested a fragment simulating projectile with an initial impact velocity of 500 m/s, which was similar to the performance of the monolithic panel.

## 1. Introduction

Future body armour materials and assemblies must have multi-functional, lightweight, and high-performance characteristics, including energy absorption, blast mitigation, and bullet/fragment/stab penetration resistance. Biological structures have evolved ingenious armour designs that inspire modern armour materials and structures [1,2,3,4,5,6]. Recently, the design and fabrication of bio-inspired materials for future military applications have attracted the attention of researchers and engineers in defence research organisations [7,8,9,10,11,12,13,14,15,16,17,18,19,20,21,22,23].

Limited studies have been conducted to assess the ballistic performance of biological structures, and marine armour systems have been the main focus. Grujicic et al. [17,18] conducted finite element (FE) simulations on a nacre-like B_4_C/polyurea staggered composite panel to investigate its performance under high-speed impact (600 m/s). The nacre-like composite significantly outperformed an equivalent monolithic ceramic panel under both normal and high-velocity conditions (31 m/s vs. 302 m/s residual velocity). Gu et al. [22] mimicked the second-order lamellar structure of a conch shell by developing a small panel using multi-material polyjet 3D printing and testing the prototype under low-velocity impact (2.3 to 3 ms^−1^). Compared to an equivalent monolithic panel made from the same stiff material, the conch-like design arrested the projectile. Abir et al. [24] developed different numerical models of panels with cross-ply, quasi-isotropic, and helicoidal laminates, respectively, which were generated based on the architecture of a crustacean exoskeleton. They investigated the performance of the selected panels under high-velocity impact from a hemi-spherical projectile. The cohesive elements were embedded in each ply to simulate matrix damage. Delamination and a through-thickness spiralling fracture were identified to be the main modes of energy dissipation. They reported a ballistic limit velocity of 151 m/s for the helicoidal composite, which was up to 8% and 15% higher than that of the cross-ply and quasi-isotropic composites, respectively.

An in-depth review of the responses of biological composite structures under dynamic loading conditions was provided in our previous paper [25]. The biological composite structures considered in this paper were chosen because they have better damage mitigation mechanisms than monolithic structures when subjected to a dynamic impact such as that from a ballistic/fragment projectile [25]. These selected biological structures generally comprise multi-layer armour systems, typically with a hard front layer to provide localised indentation resistance and a soft backing layer to dissipate energy and distribute damage. The key features of these selected biological structures in the context of ballistic impact are summarised below:Nacre from mollusc shells: A two-layer ceramic armour system is observed at the macroscale, with a staggered, multi-layer, polygonal tablet structure at the microscale (Figure 1b). The function of the hard outer layer is to resist perforation, and the nacreous backing layer dissipates energy via crack deflection and bridging.Conch (Strombus gigas): Cross-lamellar ceramic armour system that dissipates energy by deflecting cracks through the plywood-like interlocking ceramic planks in the microstructure (Figure 1c), which are oriented at around 45° in the middle layer.Fish scales (Arapaima gigas): Flexible multi-layer armour system which consists of a hard bony hydroxyapatite (mineral) macro-layer to resist perforation, with orthogonal, plywood-like, fibre-reinforced backing layers at the microscale (Figure 1d) to dissipate energy.Crustacean (Homarus americanus) exoskeleton: Helicoidal fibre-reinforced structure which comprises a dense frontal layer (exocuticle) with high hardness and a more porous rear layer (endocuticle) for absorbing energy from impacts. These layers have a twisted, plywood-like microstructure (Figure 1e) to deflect cracks.

## 2. Finite Element Model Development for Bio-Inspired Structures

### 2.1. Generating the Bio-Inspired Numerical Model

The purpose of this section is to provide an overview of the computational framework (see Figure 2) which demonstrates the general process for generating the finite element model for the different bio-inspired architectures. It is worth noting that this framework was developed to be general, and it can thereby be adapted to generate finite element models for all the bio-inspired architectures considered in this study. The framework exploits already existing functionalities in commercial programs, such as finite element meshing, and employs algorithms to provide additional functionality for geometry and mesh operations. Figure 3a,b illustrate the views of a 3D meshed nacre-like layer and its cohesive element configuration, respectively. Layer waviness can be introduced using a sinusoidal function, as shown in Figure 3c. The code is not presented due to its simplicity.

The bio-inspired finite element (FE) models were generated to mimic the structures of nacre and conch and are illustrated in Figure 4. Each panel had overall dimensions of 100 × 100 × 4.5 mm. The strike face was composed of a 1 mm thick monolithic ceramic plate and a 1.5 mm thick bio-inspired component, and the Kevlar composite backing had a thickness of 2 mm. To translate the biological architectures in the FE model, key features of the nacre, conch, fish scales, and crustacean were selected with the goal of maintaining indentation resistance at the localised impact zone whilst distributing the energy dissipation mechanisms (including crack deflection and bridging) through the rear layers. This level of simplification was based on the activation of these well-known damage tolerance mechanisms in composite structures. The bio-inspired component was composed of six layers with a thickness of 0.25 mm. The mesh dimensions of the entire panel were 0.5 mm (in-plane) and 0.25 mm (through the thickness). The intra-layer adhesive bonds were modelled with zero-thickness cohesive elements.

### 2.2. Material Models

The Johnson–Holmquist damage model was used to capture the damage in the ceramic (silicon carbide) hard phase, which has been widely used for predicting the rate-dependent behaviour of ceramics under ballistic impact. A detailed description of the model and material properties is provided in [26,27]. A traction–separation law was used to capture delamination in the inter-tablet and interlayer adhesives (vinyl ester material). A detailed description of the model and material properties is provided in [28], which investigated the behaviour of a nacre-like panel under impulsive loading. A linear transversely isotropic constitutive law was assigned to the Kevlar 129 composite material. Hashin’s model was applied to investigate the damage initiation and material failure of the homogenised Kevlar material. A detailed description of the model is provided in [29]. The material properties of the Kevlar composite are provided in Table 1.

A rate-dependent material model was adopted to simulate the behaviour of the Steel 4310 FSP. The Johnson–Cook constitutive law for ductile metals [31] was utilised to predict this behaviour, as follows:(1)$$\sigma =\left[A+B\epsilon^{n}\right]\left[1+Cln{\dot{\epsilon }}^{*}\right]\left[1-T^{*m}\right]$$
where *A* represents the yield stress; ϵ is the equivalent plastic strain; *B* and *n* account for the effects of strain hardening; ${\dot{\epsilon }}^{*}=\dot{\epsilon }/{\dot{\epsilon }}_{0}$ is the dimensionless strain rate for the reference strain rate ${\dot{\epsilon }}_{0}=0.001s^{-1}$; the constant *C* is obtained from the experiment (tension, torsion, etc.); and the temperature $T^{*m}$ is ignored, assuming isothermal conditions. The material properties for Steel 4310 are listed in Table 2. The Steel 4310 alloy was chosen because it is representative of the material of a 0.22 calibre fragment simulating projectile (FSP).

As shown in Figure 4, the panels were subjected to impact by a 0.22 calibre type 1 fragment simulating projectile (FSP) at a velocity of 500 m/s. This calibre was chosen because it is representative of fragments generated when a bomb, grenade, or artillery shell explodes on the battlefield during a military conflict (see NATO Standardization Agreement (STANAG 2920) on the ballistic test method for personal armour materials and combat clothing). The panels were clamped on all four edges. The architectures of the nacre-like and conch-like panels, along with the plain layered panel, are illustrated in Figure 4c. The performances of these panels, which were used as the baseline cases, were benchmarked against the performance of an equivalent monolithic panel using the residual velocity of the projectile and the damage distribution (to assess their multi-hit capability) as the key performance criteria. The equivalent monolithic panel is composed of a 2.5 mm thick monolithic ceramic plate and a 2 mm thick Kevlar composite backing plate. The interaction between the ceramic strike face components, Kevlar backing, and FSP were simulated using the hard contact algorithm in ABAQUS with a friction coefficient of 0.3. Tie constraints were used to connect the ceramic layers and Kevlar backing to the adhesive.

### 2.3. Bio-Inspired Armor Panel Configurations

An extensive numerical study was conducted using the present models to investigate the effects of the bio-inspired armour geometries, FSP impact velocity, and material properties on their ballistic protection performance and dynamic impact response, including the residual velocity of the FSP, the plastic/damage dissipation energy, and the damage distribution in the bio-inspired armours under ballistic impact. For the nacre-inspired composite panels, the geometrical parameters considered included the width of the tablets ($w_{t}):$ 2.5, 5, and 10 mm; the bio-inspired layer thickness (t_l_): 31.25, 62.5, 125, and 250 μm; the number of interface waves (n): 0, 10, 20, and 30; and the type of Kevlar composite backing plate: monolithic Kevlar plate or Kevlar laminate. The FSP impact velocity was chosen to be 500, 750, and 1000 m/s. For the conch-inspired composite panels, the geometrical parameters considered included the width of the planks ($w_{p}):2,\;5\;\mathrm{and}\;10\;\mathrm{mm}$ and the cross-lamellar out-of-plane angle (θ):45°, 60°, and 75°. A comparison of the dynamic response between the nacre-like and hybrid nacre/conch-like panels under FSP impact was carried out. The selected fish scales and crustacean bio-inspired structures were also studied to investigate the effects of different bio-inspired architectures on the ballistic performance of the panels with the same dimensions and component material properties. For the fish scale-inspired composite panels, the plank angle was kept constant at 90°, and the width of the planks was maintained at 10 mm. For the crustacean-inspired panels, the helical pitch angle between the planks was kept constant at 20°, and the width of the planks was also maintained at 10 mm. The composite strike face was modelled using C3D8R solid brick elements (hard ceramic component) and COH3D8 zero-thickness cohesive elements (soft adhesive component), and the Kevlar backing was modelled using SC8R continuum shell elements. The FSP bullet was modelled using C3D8R solid brick elements. The mesh dimensions of the panels considered were selected to be 0.5 mm (in-plane) and 0.25 mm (through the thickness); these were determined based on the mesh convergence study below. The FSP elements had the dimensions of 1 × 1 × 1 mm.

## 3. Numerical Study Results and Discussions

### 3.1. Influence of Nacre-like Architecture

The baseline architecture of the flat nacre-like panel shown in Figure 4c was built from several alternating layers of tablets (or bricks), the geometries of which are shown in Figure 5a,b. This structure was chosen to represent the staggered multi-layer tablet microstructure of nacre to activate the energy dissipation mechanisms of crack deflection and bridging (as discussed in Section 2.1) in the event that the projectile penetrated the hard front layer of the armour panel. These layers were staggered and stacked on top of one another to form a panelised composite structure. The layers were bonded together using a ductile adhesive. Figure 4a above illustrates the FE model for a bio-inspired panel composed of 1 mm thick monolithic SiC plate; a 1.5 mm thick nacre-like plate with six layers composed of SiC tablets and vinyl ester; and a 2 mm thick Kevlar composite backing plate. The configuration of the tablets in the 1st, 3rd, and 5th layers of the nacre-like plate is shown in Figure 5a and that in the 2nd, 4th, and 6th layers is shown in Figure 5b. Each tablet had dimensions of 10 × 10 mm. The stacking of the tablets through the thickness of the panel is shown in Figure 5c.

A mesh convergence study was conducted using the FSP velocity and the plastic dissipation energy in the nacre-like strike face. The element size was investigated in the nacre-like layer as this is the main focus of this work. It can be observed in Figure 6 that the models with element sizes of 0.5 mm and 0.25 mm show a highly similar FSP velocity (Figure 6a) and plastic dissipation energy (Figure 6b). The 1 mm model terminated prematurely due to excessive element distortions. Therefore, an element size of 0.5 mm was selected to model all the panels simulated in this work. The performance of the nacre-like panel was compared to that of the equivalent monolithic panel by assessing the residual velocity of the FSP at different incident velocities, namely 500, 750, and 1000 m/s. In this comparison, the adhesive between the nacre-like layers was removed, given that the inter-layer adhesive was mainly responsible for creating weak points in the panel. It can be observed in Figure 7 that the nacre-like architecture with the removed adhesive (Figure 7a) showed the same residual velocity as that of the monolithic panel (Figure 7b). It is suspected that this similarity is attributed to the highly confined damage zone under high-velocity impact, where the fracture in the nacre-like panel is more localised in the vicinity of the projectile compared to that of the equivalent monolithic panel. The damage distribution in both panels is assessed further below to ascertain whether there are other improvements in ballistic protection, e.g., multi-hit resistance.

In this study, the tablet width ($W_{t}$) shown in Figure 8a was selected to be 2.5, 5, and 10 mm, respectively, and the FSP impact velocity was chosen to be 500 m/s. The number of layers shown in Figure 5c was chosen to be the same for all the panels considered in this subsection. Essentially, this changed the overlap length from 1.25 mm to 5 mm between the tablets in the adjacent layers. It can be observed in Figure 8b that this parameter had no obvious effect on the residual velocity of the projectile. Figure 8c illustrates the fact that the energy dissipated in the strike face was reduced with a decrease in tablet width. In contrast, the energy dissipated in the Kevlar composite component increased with a decrease in tablet width (Figure 8d). The implication is that a smaller tablet width results in more localised damage to the Kevlar composite backing plate and thereby could inhibit the ballistic performance of the panel.

In this study, the thickness of a layer in the nacre-like component (i.e., $t_{l}$ shown in Figure 9a) was varied whilst the overall thickness of the nacre panel was kept constant. The FSP impact velocity was chosen to be 500 m/s. It can be observed in Figure 9b that the FSP residual velocity decreased when the layer thickness was reduced. This behaviour indicates that crack deflection and bridging through the thickness, which are common in laminated composites, may arrest the projectile. It is evident that a layer thickness of 62.5 μm, which equates to 24 layers, significantly leads to a reduction in the residual velocity of the projectile. It can be observed in Figure 9c,d that there was a prominent increase in the plastic dissipation energy in the strike face and a reduction in the damage dissipation energy in the Kevlar composite backing plate with a decrease in layer thickness. This implies that the panel with a lower value of $t_{l}$ may absorb more impact energy and receive less damage in the Kevlar composite backing plate.

Figure 10 illustrates the damage zones at the strike face of the flat nacre-like panel and monolithic ceramic composite panel, respectively. It can be observed that the damage zone in the vicinity of the projectile was prominently more localised in the nacre-like panel (area of $650\;{\mathrm{mm}}^{2}$) compared to that in the monolithic panel (area of $1375\;{\mathrm{mm}}^{2}$), which underwent catastrophic failure. This indicates that the nacre-like panel possesses better multi-hit capabilities.

In this study, the number of waves (n in Figure 11a) on the surface and at the interface of the wavy nacre-like panels under impact from an FSP with an initial impact velocity of 500 m/s was selected to be 10, 20, and 30, respectively. Although it is evident in Figure 11b that the projectile was not arrested for all cases and that the influence of n on the FSP residual velocity was not significant, it can be observed in Figure 12 that the waviness with a low number of waves (n) resulted in the oblique impact of the projectile and thereby rotated it. It can be deduced from Figure 11c,d that an increase in the number of waves (n) did not show a monotonic trend in the energy dissipated by the strike face and the Kevlar composite backing components. For the cases considered, the highest number of waves (n=30) showed the highest plastic dissipation energy/damage to the strike face (Figure 11c), whereas the lowest number of waves showed the highest damage in the Kevlar composite backing (Figure 11d). Effectively, increasing the number of waves any further would result in a relatively flat strike face.

### 3.2. Influence of Conch Architecture

The conch-like panel was modelled by skewing the cohesive bonds in the middle layer by an angle θ to mimic the cross-lamellar structure observed in the macro-layers of the conch shell (Figure 13a), namely the external, cross-lamellar, and internal layers. The front layer of the panel (monolithic ceramic layer in Figure 13a) was modelled using the same architecture of nacre. The channel cracks in the front layer were expected to propagate through the external conch-like layer and to then be deflected by the middle cross-lamellar layer. The back (internal) layer was modelled with thin monolithic components stacked on top of one another, whereby delamination was expected to dissipate energy from the projectile. The width of the planks ($w_{p}$) was also varied to assess the panel’s performance.

It can be observed in Figure 13b that reducing the width of the planks almost did not affect the residual velocity of the FSP. However, the energy dissipation decreased in the strike face (Figure 13c) and remained constant in the Kevlar backing plate (Figure 13d). For the conch-like panels with plank widths of 2.5 and 5 mm, the simulation was terminated (see Figure 13b–d) due to the higher amount of adhesive in the impact region.

In this study, the angle of the cohesive bonds between the planks in the middle layer (θ) (Figure 14a) was varied to determine whether the projectile rotated. For the cases considered, the thickness of the plank in the middle layer was chosen to be 0.5 mm. It can be observed in Figure 14b that the residual velocity of the FSP was reduced from 200 m/s (θ=45° or 60°) to 150 m/s (θ=75°), indicating that projectile rotation occurred. In terms of ballistic protection performance, a higher skew angle resulted in lower plastic dissipation energy/damage to the strike face (Figure 14c) but higher damage in the Kevlar backing plate (Figure 14d). This behaviour is attributed to the better distribution of damage away from the impact zone, which was due to the crack deflection produced by the oblique bond angles. Consequently, the damage was distributed to more material in the Kevlar backing layer, and this increased energy dissipation resulted in a lower residual velocity of the projectile.

### 3.3. Influence of Hybrid Nacre/Conch Architecture

In this study, the nacre-like and conch-like architectures were combined to generate a hybrid nacre/conch-like panel, in which the thickness of the nacre-like or conch-like component was chosen to be 62.5 μm, and the cross-lamellar out-of-plane angle (θ) was chosen to be θ=75° (Figure 15a). The reason why this architecture was investigated was attributed to a study which showed that the thinner layers in the nacre-like layer and the higher value of the cross-lamellar bond angle in the conch-like layer resulted in a higher reduction in the velocity of the FSP compared to the other investigated panels. It was observed in Figure 15b that the hybrid structure had a higher FSP residual velocity compared to the nacre-like panel. Furthermore, excessive element distortions occurred in the impact zone due to the instabilities caused by a large value of θ, which terminated the simulation prematurely. Hence, for the cases considered previously, the nacre-like architecture is the most effective thus far in reducing the residual velocity of the FSP, as well as the size of the damage zone in the vicinity of the impact region.

### 3.4. Influence of Plain Layered Architecture

In this study, plain monolithic ceramic panels were stacked on top of one another, and the adjacent layers were bonded by adhesives, as shown in Figure 16a. As with the nacre-like panels, an increase in the layer thickness resulted in an increase in the FSP residual velocity (Figure 16b). A comparison between the nacre-like panel (Figure 9b) and the plain layered panel (Figure 16b), which had the same layer thickness of 62.5 μm, shows that the FSP residual velocity for the nacre-like panel (80 m/s) was much lower than that of the plain layered panel (150 m/s). Moreover, a reduction in $t_{l}$ led to an increase in the energy dissipation in the strike face (Figure 16c) and a decrease in the damage dissipation energy in the Kevlar composite backing (Figure 16d). However, the Kevlar composite backing plate in the plain layered panel showed excessive element distortions in the impact zone (terminated simulation) due to localised large deformations ahead of the projectile.

### 3.5. Ballistic Performance Benefits of Bio-Inspired Panels

In this section, several studies are conducted to compare the performance of different biomimetic architectures under high-speed impact from an FSP. The performance measures are projectile rotation and residual velocity, and damage localisation and area. Comparisons of the FSP residual velocity, damage area, and FSP rotation between the nacre-like, conch-inspired, plain layered monolithic and plain monolithic ceramic panels are illustrated in Figure 17 and Figure 18, respectively, in which all panels considered have dimensions of 100 × 100 × 4.5 mm. It can be inferred from Figure 17 that the monolithic panel arrested the projectile, and the nacre-like architecture with $t_{l}=62.5\;\mu m$ resulted in the second-best ballistic performance. However, according to the damage patterns in Figure 18, the damaged area in the monolithic panel was much larger, which resulted in lower multi-hit resistance. In contrast, the nacre-like and conch-like architectures showed prominently smaller damage zones (see Figure 18). Zooming in on the damage zones in Figure 18, the nacre-like and conch-like architectures had the effect of destabilising the projectile, i.e., rotation can be observed (see Figure 18a,b), whereas rotation of the projectile was not obvious for the plain layered monolithic and plain monolithic ceramic panels (see Figure 18c,d). This rotation occurs due to non-uniform contact between the projectile and damaged area of the panel, which effectively induces a torque that rotates it.

The aim of this study is to determine the influence of the inter-layer orientation on the ballistic resistance of the selected bio-inspired panels. In each case, the thickness of the interlayer adhesives was chosen to be 100 μm. Given the restrictions in the thickness of the armour panel, several inter-layer orientations were generated: conch-like cross-plywood (alternating ±45° plies); fish scale-inspired orthogonal plywood (alternating 90° plys), and crustacean exoskeleton Bouligand helical structures (plies pitched at 20° between layers). These orientations were based on the different bio-inspired architectures explained in Section 1 (schematically shown in Figure 1 and Figure 4), with the main goal of dissipating energy from the projectile via crack deflection. A comparison of the FSP velocity between the selected bio-inspired panels bonded with and without adhesive was also conducted.

It can be observed in Figure 19a that for the cases considered, the panels with adhesives between the layers failed to stop the projectile, and the residual velocity profiles were almost the same. Hence, no further parametric studies were conducted on these architectures. Figure 19b shows the residual velocity profiles for the panels where the layers were assumed to be perfectly bonded without adhesive, and they were also similar. The FSP was stopped by all the panels considered. Hence, under high-velocity impact, the architectures for the selected panels appeared to have no effect on the residual projectile velocity, but the damage zone may be confined such that the multi-hit resistant capability of the panel might be improved. To this effect, the damage patterns in each panel are assessed further below.

Figure 20 shows that the damage shapes on the front surface of the monolithic ceramic plate were almost identical for all of the bio-inspired panels considered, and the damage area in the crustacean-like panel was slightly larger than the others. The damage on the front surface of the bio-inspired layer behind the monolithic ceramic plate appears to be more confined to the impact zone for the nacre-like panel due to the staggered platelet architecture, which indicates that the nacre-like panel has better multi-hit resistant capability than the others. The most prominent result is observed on the front surface of the rear layer of the panel (Figure 20), in which the damage in the nacre-like panel was significantly more confined to the impact zone. This demonstrates its superior multi-hit capability compared to the other bio-inspired panels.

## 4. Conclusions

A finite element modelling framework was developed to mimic several biological composite architectures that have the potential to defeat ballistic impact, including those of nacre, conch, fish scales, and the crustacean exoskeleton. Studies were conducted to assess the performance of the bio-inspired panels under ballistic impact from a fragment simulating projectile. The key performance criteria were the residual velocity of the projectile and the multi-hit resistant capability (area of the damage zone). The most prominent findings from this study are summarised as follows:In general, the multi-layer bio-inspired panels with inter-layer adhesives resulted in a confined damage zone, whereas the monolithic panel underwent catastrophic failure. This was attributed to lower stiffness and crack bridging in the bio-inspired panels.The nacre-like and conch-like architectures possessed better multi-hit capabilities as they confined the damage zone in the vicinity of the projectile. Asymmetry in the staggered structure resulted in some rotation of the projectile when it penetrated into the panel.By removing the inter-layer adhesives, introducing cuts into a monolithic panel directly, and filling those with an adhesive material, all of the bio-inspired architectures considered arrested the projectile. The nacre architecture showed the most confined damage zone.The thickness of the nacre-like tablet layers had a significant effect on the ballistic performance of the panel, reducing the residual velocity by more than 50% when the thickness decreased. In contrast, the overlap length between the tablets had no impact on ballistic performance.For the conch-like panel, a higher lamellar angle reduced the residual velocity by 25%, whereas the plank width had a marginal effect on ballistic performance.The nacre-like panel with the lowest layer thickness (more tablet layers) was significantly more effective than the hybrid nacre/conch architecture in reducing the residual velocity of the projectile due to crack bridging effects. This nacre-like panel was also more effective than the plain layered panel with the same layer thickness.For the considered bio-inspired panels with and without adhesives between the layers, the effect of architecture on the residual projectile velocity was not significant, but the damage zone was more confined. Therefore, the multi-hit resistant capability of the panel was sensitive to the architecture.

## Figures and Tables

**Figure 1 biomimetics-08-00195-f001:**
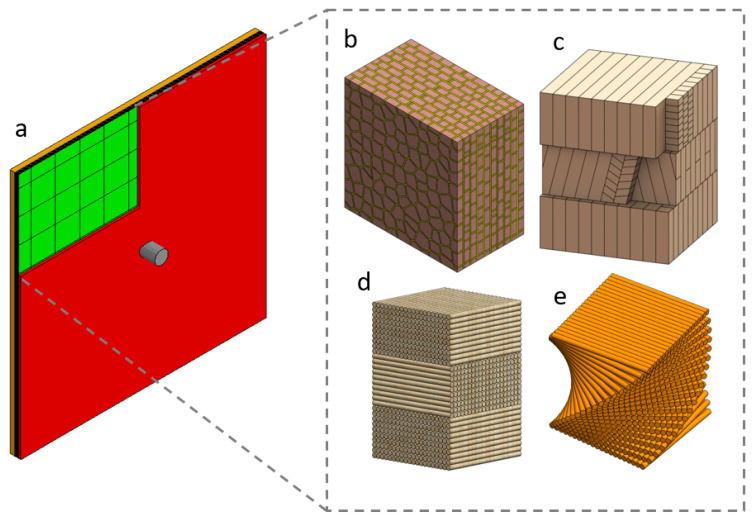
Schematics of (**a**) panel under ballistic impact from a fragment simulating projectile (FSP); (**b**) nacre’s staggered tablet microstructure; (**c**) plywood-like microstructure in a conch shell; (**d**) orthogonal fibre-reinforced structure in fish scales; and (**e**) twisted fibre structure in a crustacean exoskeleton [25].

**Figure 2 biomimetics-08-00195-f002:**
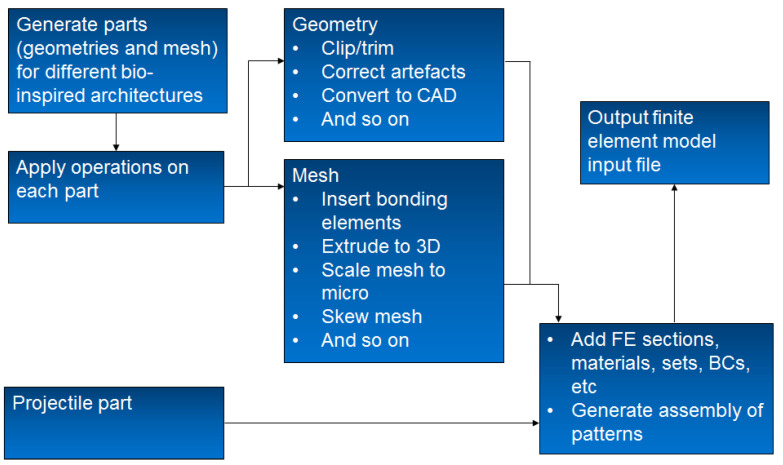
Schematic of computational framework for generating bio-inspired FE models.

**Figure 3 biomimetics-08-00195-f003:**
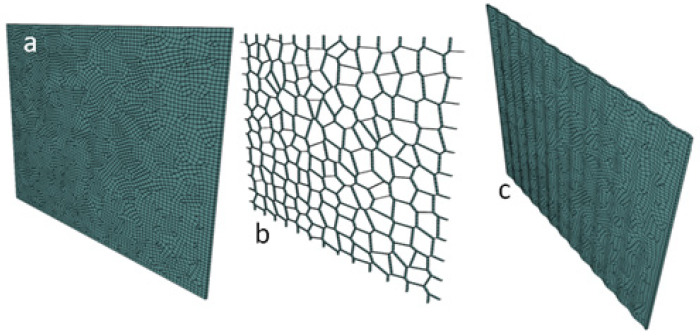
(**a**) meshed nacre-like layer; (**b**) cohesive element insertion; and (**c**) wavy tablet profile generated by applying a sinusoidal function.

**Figure 4 biomimetics-08-00195-f004:**
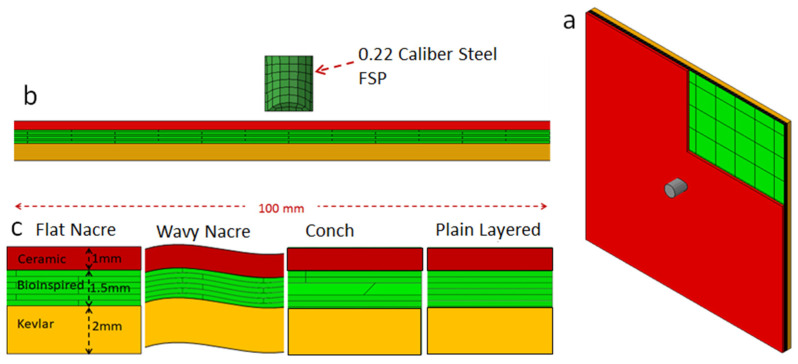
Initial baseline composite armour panels based on the architectures of nacre and conch, which are composed of a thin monolithic plate, a bio-inspired middle layer, and Kevlar backing: (**a**) 3D model; (**b**) elevation view; and (**c**) zoomed in elevation view.

**Figure 5 biomimetics-08-00195-f005:**
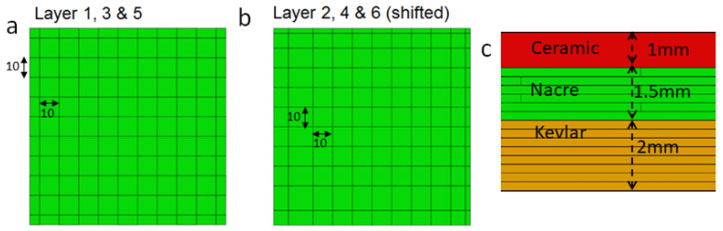
Flat nacre-like architecture, which consists of square tablets bonded by a thin adhesive: (**a**) layers 1,3, and 5; (**b**) layers 2,4, and 6; and (**c**) through thickness.

**Figure 6 biomimetics-08-00195-f006:**
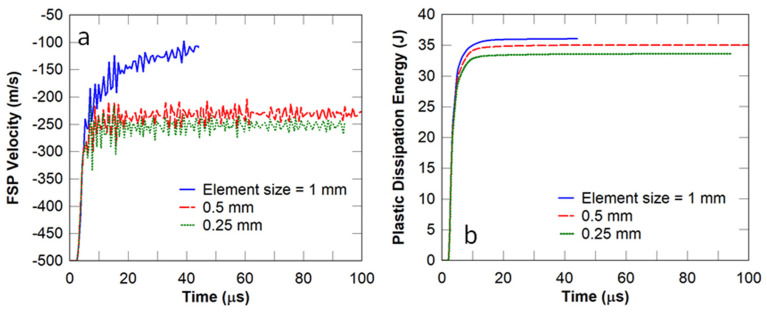
Mesh convergence study: (**a**) fragment simulating projectile velocity and (**b**) plastic dissipation energy in the flat nacre−like strike face.

**Figure 7 biomimetics-08-00195-f007:**
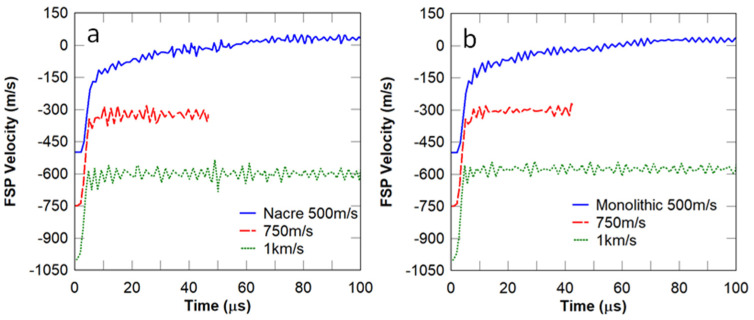
Baseline flat nacre−like architecture compared to monolithic panel at different incident velocities: (**a**) flat nacre−like and (**b**) monolithic panel. In this case, there is no inter−layer adhesive, i.e., the tablet layers are tied together between layers.

**Figure 8 biomimetics-08-00195-f008:**
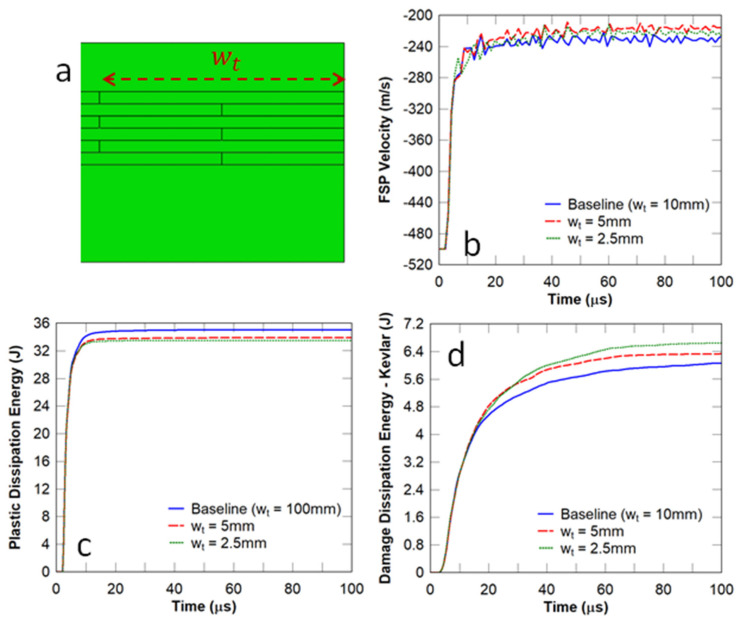
Influence of the tablet width on the dynamic responses of flat nacre−inspired panels under impact from an FSP with an initial velocity of 500 m/s: (**a**) tablet width; (**b**) FSP velocity; (**c**) plastic dissipation energy in the strike face; and (**d**) damage dissipation energy in the Kevlar composite backing plate.

**Figure 9 biomimetics-08-00195-f009:**
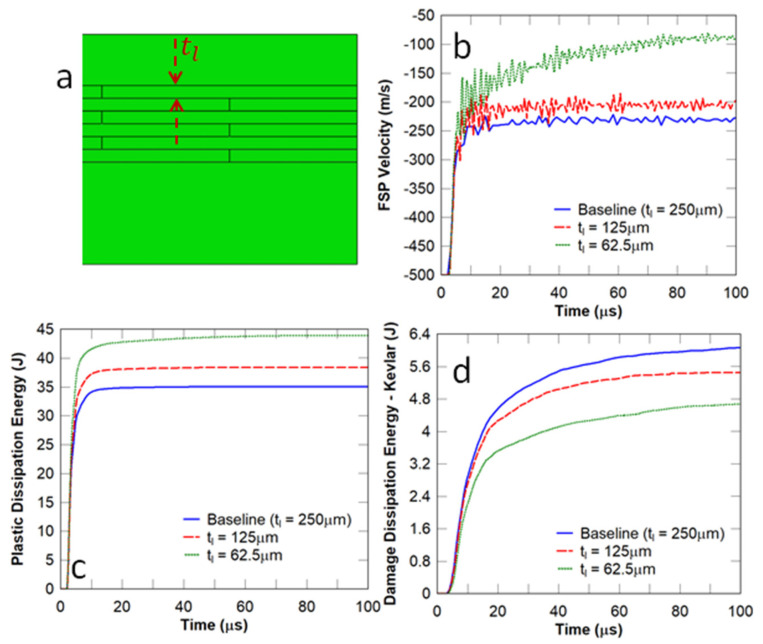
Influence of the layer thickness on the dynamic responses of flat nacre−inspired panels under impact from an FSP with an initial velocity of 500 m/s: (**a**) layer thickness; (**b**) FSP velocity; (**c**) plastic dissipation energy in the strike face; and (**d**) damage dissipation energy in the Kevlar backing plate.

**Figure 10 biomimetics-08-00195-f010:**
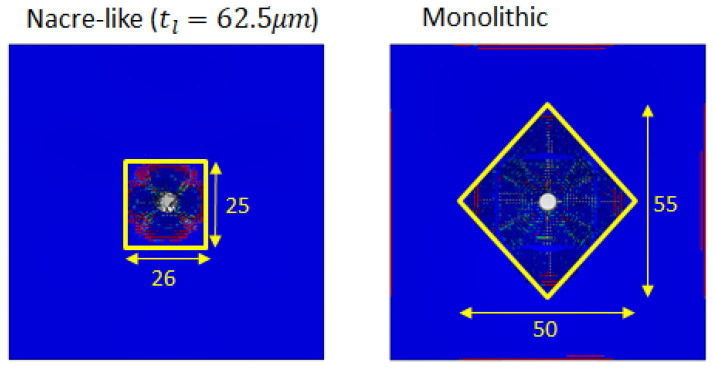
Damage zones in the flat nacre-like panel ($650\;{\mathrm{mm}}^{2}$) and the equivalent monolithic panel ($1375\;{\mathrm{mm}}^{2}$) at the strike face under impact from an FSP with an initial velocity of 500 m/s.

**Figure 11 biomimetics-08-00195-f011:**
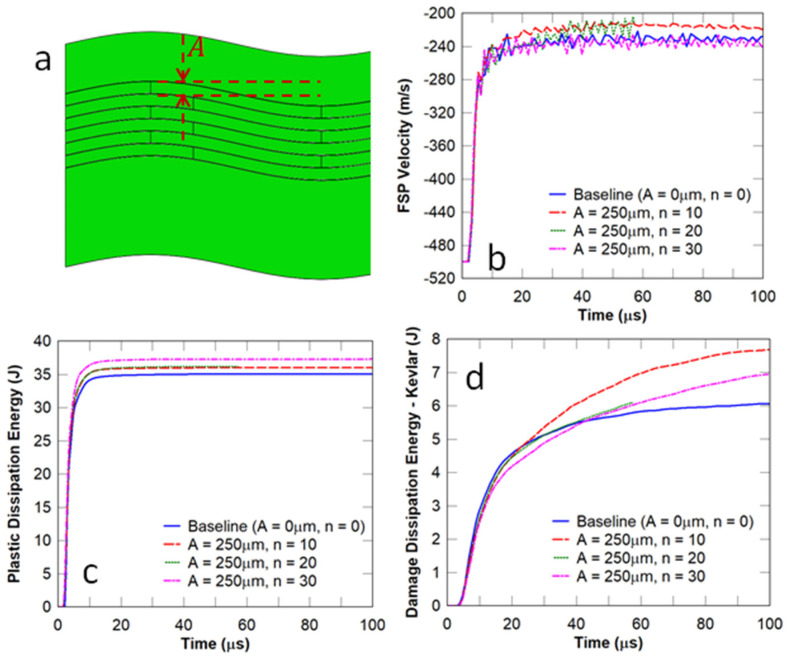
Influence of the waviness (number of waves or wavelength): (**a**) wavy tablets; A is the waviness amplitude; (**b**) FSP velocity; (**c**) plastic dissipation energy in the strike face; and (**d**) damage dissipation energy in the Kevlar backing plate.

**Figure 12 biomimetics-08-00195-f012:**
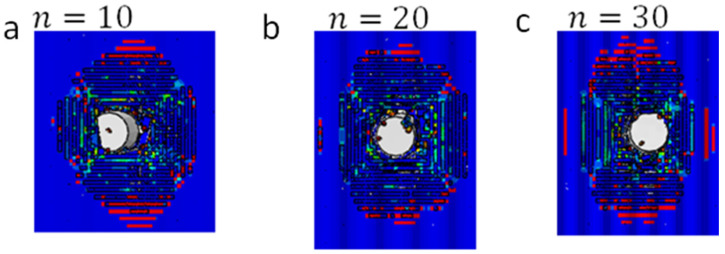
FSP rotation in the wavy nacre−like panels with different numbers of waves: (**a**) 10 waves; (**b**) 20 waves; and (**c**) 30 waves.

**Figure 13 biomimetics-08-00195-f013:**
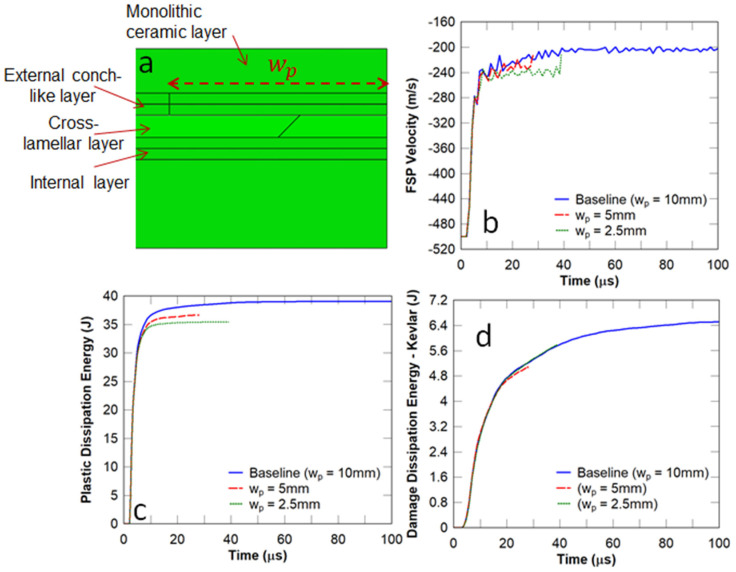
Conch−like architecture—influence of plank width: (**a**) conch−like panel; (**b**) FSP velocity; (**c**) plastic dissipation energy in the strike face; and (**d**) damage dissipation energy in the Kevlar composite backing plate.

**Figure 14 biomimetics-08-00195-f014:**
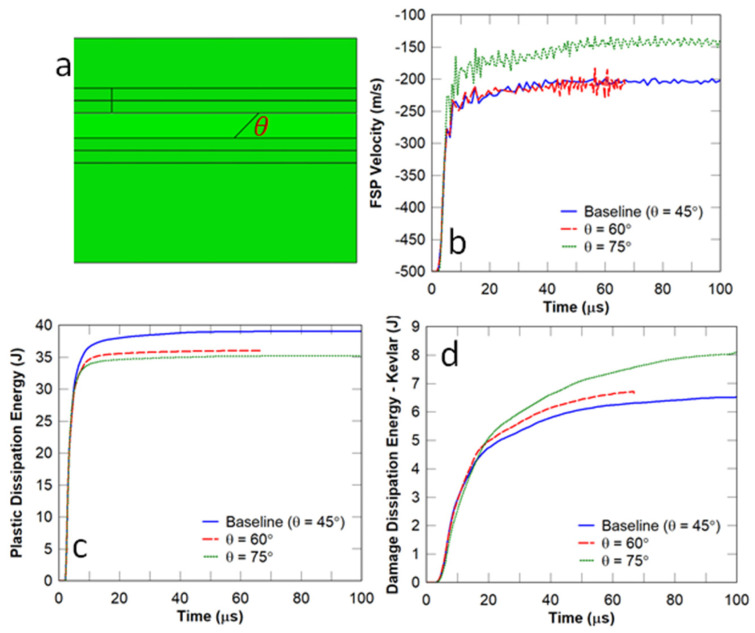
Conch-like architecture—influence of cross-lamellar out-of-plane angle: (**a**) conch-like panel; (**b**) FSP velocity; (**c**) plastic dissipation energy in the strike face; and (**d**) damage dissipation energy in the Kevlar backing plate.

**Figure 15 biomimetics-08-00195-f015:**
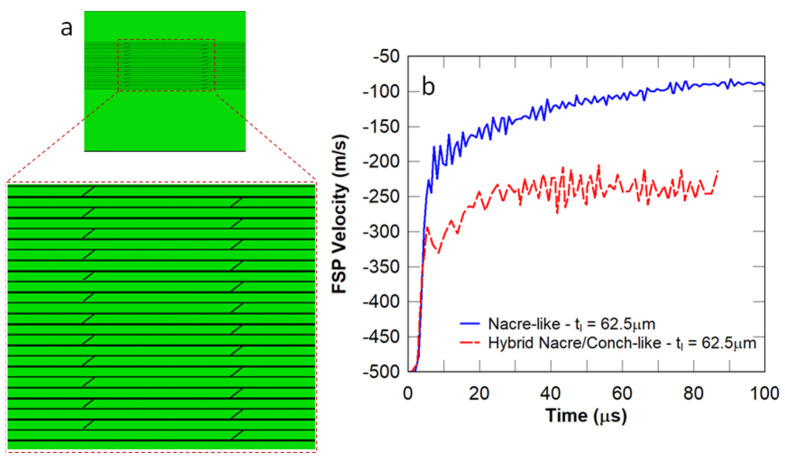
(**a**) Hybrid nacre/conch−like architecture and (**b**) FSP velocity.

**Figure 16 biomimetics-08-00195-f016:**
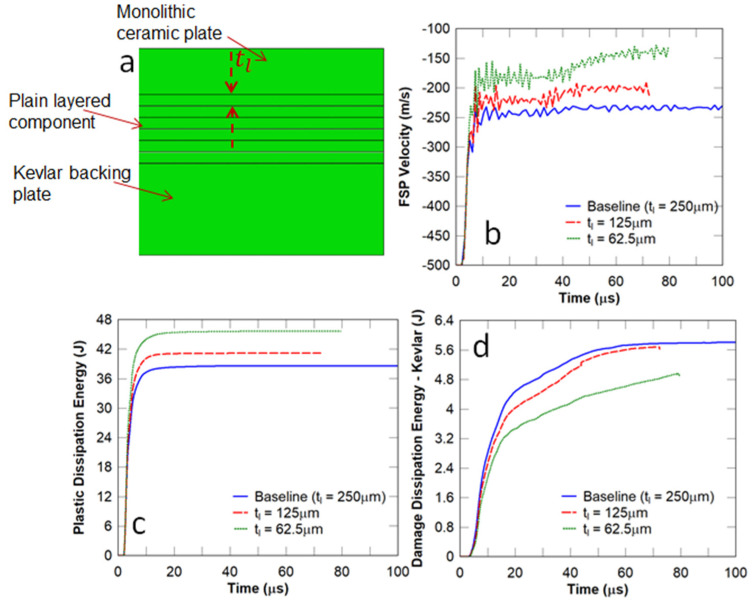
Plain layered monolithic panel—influence of the layer thickness: (**a**) plain layered panel; (**b**) FSP velocity; (**c**) plastic dissipation energy in the strike face; and (**d**) damage dissipation energy in the Kevlar backing plate.

**Figure 17 biomimetics-08-00195-f017:**
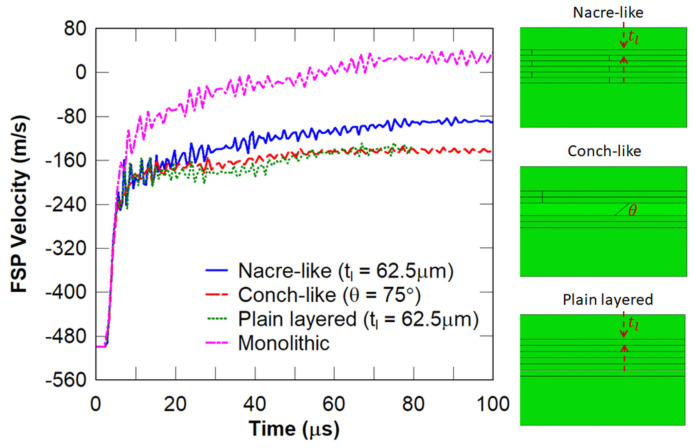
Variations of the predicted FSP velocity for the selected panels.

**Figure 18 biomimetics-08-00195-f018:**
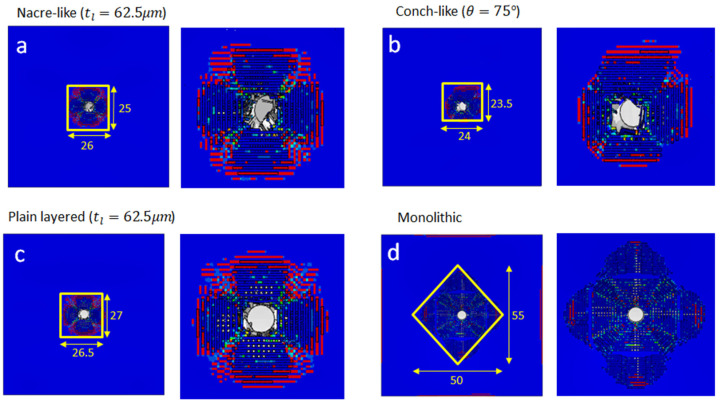
Damage zones in the strike faces of: (**a**) nacre-like ($650\;{\mathrm{mm}}^{2}$); (**b**) conch-like ($564\;{\mathrm{mm}}^{2}$); (**c**) plain layered ($715.5\;{\mathrm{mm}}^{2}$); and (**d**) monolithic ($1375\;{\mathrm{mm}}^{2}$) panels. The zoomed-in damage zones show projectile rotation.

**Figure 19 biomimetics-08-00195-f019:**
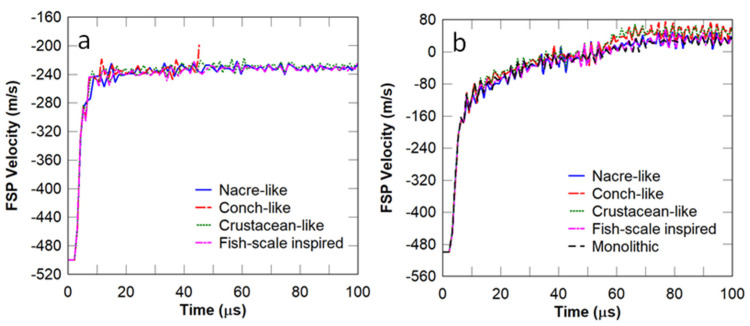
FSP velocity for the different bio−inspired panels: (**a**) with adhesives between layers and (**b**) without adhesives between layers.

**Figure 20 biomimetics-08-00195-f020:**
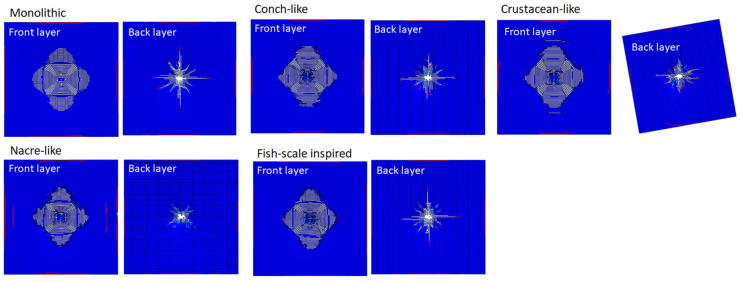
Different bio-inspired architectures: damage shapes in the monolithic ceramic panel and the selected bio-inspired panels (front and back layers). Note that the crustacean back layer is rotated due to the Bouligand variations in layer angles.

**Table 1 biomimetics-08-00195-t001:** Kevlar 129 properties obtained from experimental work [30], including the density (ρ), Young’s moduli in the plane of the Kevlar layer ($E_{11}$, $E_{22}$), in-plane shear modulus ($G_{12}$), in-plane tensile ($X^{T}$,$Y^{T}$) and compressive strengths $(X^{C}$,$Y^{C}$), and shear strength ($S^{T}$).

Property	Value	Material Property	Value
Density, *ρ*	1230 kg/m^3^	$X^{T}$ (Tensile)	800 MPa
$E_{11}$	22 GPa	$X^{C}$ (Compressive)	800 MPa
$E_{22}$	22 GPa	$Y^{T}$ (Tensile)	800 MPa
$G_{12}$	770 MPa	$Y^{C}$ (Compressive)	800 MPa
Poisson’s ratio, ν	0.25	$S^{T}$ (Shear)	1000 MPa

**Table 2 biomimetics-08-00195-t002:** Steel 4310 FSP properties [32], including the density (ρ), Young’s modulus (*E*), Poisson’s ratio (ν), yield strength (*A*), tangent modulus (*B*), strain hardening exponent (*n*), and temperature exponent (*m*).

Property	Value	Property	Value
Density, *ρ*	7833.4 kg/m^3^	Plastic	
Elastic		*A*	1076.7 MPa
*E*	206,840 MPa	*B*	5703.1 MPa
*ν*	0.29	*n*	0.276266
		*m*	0

## Data Availability

The corresponding author may be contacted for access to the data.

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
