# Peer review of "A Numerical Modelling Framework for Investigating the Ballistic Performance of Bio-Inspired Body Armours"

_biomimetics, 2023, doi:10.3390/biomimetics8020195_

Round 1

Reviewer 1 Report

The manuscript describes an ambitious approach of comparing several different, bio-inspired structures for their impact behaviour. The results could be interesting but unfortunately, little explanation is provided why the selected examples were chose and therefore the selection seems arbitrary, which level of abstraction was used when translating the biological model into the tested structure the discussion of these results is minimal and no context is provided to put the observed results in perspective.There is not a single reference cited in the discussion! Additionally some section put into

Overall the manuscript needs more explanation and discussion of the presented results and chosen methodology to better understand the significance of the presented results. 

More detailed comments below:

P3L46-59: This is a summary of the work conducted. This should not be at the end of the introduction and can either be deleted or needs to be rephrased.

P3L50: Which species of fish, conch and crustacean were selected? It is not realistic all those animals show exactly the same biological architecture, so please specify.

Section 2: Which level of simplification was used to translate the biological architecture in the FEM model? Why was the level of simplification considered suitable? The authors needs to provide a clear explanation here.

P6 Table 1: The authors needs to explain the used variables in the table header and should also provide a short explanation why Kevlar is assumed to have the same in tension and compression.

P7 Table 2: As above, explain the used variables and which property they represent. Why was this steel alloy chosen and not another one?

P7L107: Which was this caliber type chosen? Please explain.

P8L124ff: The description  of the used samples should be in the methods section, not in the discussion.

P9 Figure 5: Why was this structure chosen to represent nacre? Were other trialed? If not, why not?

P16L264-266: The authors declare "In terms of ballistic protection
performance, a higher skew angle results in lower plastic dissipation energy/damage to the strike face (
Figure 14c) but higher damage in the Kevlar backing plate (Figure 14d)." But they don't explain why and how this statement can be concluded from their results. Please provide a more detailed explanation. Is there any literature that can support this statement?

P17L278: The authors describe a reduction of velocity compared "to others". Which others? Please specify.

P18L296: "However, the Kevlar composite backing plate in the plain layered panel shows excessive element distortions in the impact zone" Why is that? Please explain. 

P19L313ff: Why is the described behaviour occurring? Explain please.

P20L325ff: Why did you choose those inter-layer orientations? Is there a reason? Was it random? Please explain.

Author Response

Thank you very much for your recommendations. The feedbacks are very encouraging and helpful for us to improve the quality of our manuscript. The responses to your comments are provided below and the revisions are provided in red text in the revised manuscript.

General comment: The manuscript describes an ambitious approach of comparing several different, bio-inspired structures for their impact behaviour. The results could be interesting but unfortunately, little explanation is provided why the selected examples were chose and therefore the selection seems arbitrary, which level of abstraction was used when translating the biological model into the tested structure the discussion of these results is minimal and no context is provided to put the observed results in perspective. There is not a single reference cited in the discussion! Additionally some section put into

Overall the manuscript needs more explanation and discussion of the presented results and chosen methodology to better understand the significance of the presented results.

Response: Thank you for your comments, which are address hereafter and help to improve the quality of our manuscript.

Comment #1: P3L46-59: This is a summary of the work conducted. This should not be at the end of the introduction and can either be deleted or needs to be rephrased.

Response #1: Thank you for your comment. The summary of the work conducted has been replaced by the prevailing features of the selected biological structures to justify why they were chosen for ballistic impact applications (lines 46-69 of the revised manuscript).

Comment #2: P3L50: Which species of fish, conch and crustacean were selected? It is not realistic all those animals show exactly the same biological architecture, so please specify.

Response #2: Thank you for your comment. The species have been specified on lines 55, 59, 62 and 66 of the revised manuscript.

Comment #3: Section 2: Which level of simplification was used to translate the biological architecture in the FEM model? Why was the level of simplification considered suitable? The authors needs to provide a clear explanation here.

Response #3: Thank you for your insightful comment. The level of simplification to translate the biological architecture in the FE model and its suitability for ballistic applications is discussed on lines 96-101 of the revised manuscript.

Comment #4: P6 Table 1: The authors needs to explain the used variables in the table header and should also provide a short explanation why Kevlar is assumed to have the same in tension and compression.

Response #4: Thank you for your comment. The variables have been explained in the header of Table 1 in the revised manuscript. We also clarified that these properties were obtained from experiments (ref #30).

Comment #5: P7 Table 2: As above, explain the used variables and which property they represent. Why was this steel alloy chosen and not another one?

Response #5: Thank you for your comment. The variables have been explained in the header of Table 2, and an explanation as to why the Steel 4310 alloy was chosen is provided on lines 128-129 of the revised manuscript.

Comment #6: P7L107: Which was this caliber type chosen? Please explain.

Response #6: Thank you for your comment. The reason as to why this caliber was chosen is provided on lines 131-134 of the revised manuscript.

Comment #7: P8L124ff: The description  of the used samples should be in the methods section, not in the discussion.

Response #7: Agreed, this information has been moved to Section 2.3 (lines 145-170) of the revised manuscript.

Comment #8: P9 Figure 5: Why was this structure chosen to represent nacre? Were other trialed? If not, why not?

Response #8: Thank you for your insightful comment. A justification for why this structure was chosen  to represent nacre is provided on lines 174-177 of the revised manuscript.

Comment #9: P16L264-266: The authors declare "In terms of ballistic protection performance, a higher skew angle results in lower plastic dissipation energy/damage to the strike face (Figure 14c) but higher damage in the Kevlar backing plate (Figure 14d)." But they don't explain why and how this statement can be concluded from their results. Please provide a more detailed explanation. Is there any literature that can support this statement?

Response #9: Thank you for your comment. A more detailed explanation is provided on lines 297-301 of the revised manuscript, which is based on current knowledge of the energy dissipation mechanisms found in composite structures.

Comment# 10: P17L278: The authors describe a reduction of velocity compared "to others". Which others? Please specify.

Response #10: Thank you for your comment. This has been clarified on line 313 of the revised manuscript.

Comment# 11: P18L296: "However, the Kevlar composite backing plate in the plain layered panel shows excessive element distortions in the impact zone" Why is that? Please explain. 

Response #11: Thank you for your comment. The reason for excessive element distortions in the impact zone has been clarified on lines 332-333 of the revised manuscript.

Comment# 12: P19L313ff: Why is the described behaviour occurring? Explain please.

Response #12: Thank you for your comment. The reason for projectile rotation has been clarified on lines 353-355 of the revised manuscript.

Comment# 13: P20L325ff: Why did you choose those inter-layer orientations? Is there a reason? Was it random? Please explain.

Response #13: Thank you for your comment. The choice of interlayer orientations has been clarified on lines 367-370 of the revised manuscript.

Reviewer 2 Report

The authors try to investigate the ballistic performance of bioinspired armors using numerical simulations. This is a good idea and the research in this field is in need. However, the study scope of this manuscript is too broad and no specific object is focused, which results in no novel or in-depth analysis at all. I was even confused that this is a research article or review paper when I was reading it. Rather than quickly dip into all four bioinspired structures and study so many influencing factors, I strongly suggest the authors could refine their research cope and provide some in-depth analysis.

Author Response

Thank you very much for your recommendations. Your comments are helpful for us to improve the quality of our manuscript. The responses to your comments are provided below and the revisions are provided in red text in the revised manuscript.

General comment: The authors try to investigate the ballistic performance of bioinspired armors using numerical simulations. This is a good idea and the research in this field is in need. However, the study scope of this manuscript is too broad and no specific object is focused, which results in no novel or in-depth analysis at all. I was even confused that this is a research article or review paper when I was reading it. Rather than quickly dip into all four bioinspired structures and study so many influencing factors, I strongly suggest the authors could refine their research cope and provide some in-depth analysis.

Response: Thank you for your general comments. The scope of the manuscript is focused on the applications of bioinspired structures in the context of ballistic impact resistance. We carefully chose specific bioinspired armour panel configurations that can provide indentation resistance in the impact zone whilst activating energy dissipation mechanisms (crack deflection and bridging) in the event that the projectile penetrates the panel. This has been clarified on lines 46-69 and 174-177 of the revised manuscript, which also include a reference to our prior detailed review of the architectures of nacre, conch, fish scale and crustacean exoskeleton in the context of impact applications (ref #25). The results discussed in Section 3 demonstrate that our postulated energy dissipation mechanisms were observed and had an influence on reducing the residual velocity of the projectile.

Round 2

Reviewer 1 Report

The authors have addressed my comments in a satisfactory manner. The manuscript can be published.